# Multi-Index Comprehensive Assessment Optimized Critical Flavonoids Extraction from *Semen Hoveniae* and Their In Vitro Digestive Behavior Evaluation

**DOI:** 10.3390/foods12040773

**Published:** 2023-02-10

**Authors:** Xiaomei Fu, Yan Tan, Meng Shi, Chaoxi Zeng, Si Qin

**Affiliations:** College of Food Science and Technology, Hunan Agricultural University, Changsha 410128, China

**Keywords:** multi-index comprehensive assessment, AHP, flavonoids, *Semen Hoveniae*, in vitro digestion, antioxidant

## Abstract

Critical flavonoids from *Semen Hoveniae* have huge potential bioactivities on hypoglycemic. A multi-index comprehensive assessment based on Analytic Hierarchy Process (AHP) method was performed to optimize the extraction process of flavonoids from *Semen Hoveniae*, which taking dihydromyricetin, taxifolin, myricetin and quercetin as indexes, and, then, an in vitro simulated gastrointestinal digestion model was established to investigate the changes of flavonoids contents and their antioxidant capacity before and after digestion. The results showed that three influence factors acted significantly with the order of ethanol concentration > solid-liquid ratio > ultrasound time. The optimized extraction parameters were as follows: 1:37 *w*/*v* of solid-liquid ratio, 68% of ethanol concentration and 45 min for ultrasonic time. During in vitro digestion, the order of remaining ratio of four flavonoids in the extract was dihydromyricetin > taxifolin > myricetin > quercetin in gastric digestion, and remaining ratio of taxifolin was 34.87% while others were restructured in intestinal digestion. Furthermore, the 1,1-dipheny-2-picryhydrazyl free radical (DPPH ·) scavenging ability and oxygen radical absorption capacity (ORAC) of extract were more stable in gastric digestion. After an hour’s intestinal digestion, the extract had no DPPH antioxidant capacity, but amazingly, its ORAC antioxidant capacity was retained or increased, which implied that substances were transformed and more hydrogen donors were produced. This study has carried out a preliminary discussion from the perspective of extraction and put forward a new research idea, to improve the in vivo bioavailability of the critical flavonoids from *Semen Hoveniae*.

## 1. Introduction

*Semen Hoveniae* is seed of *Hovenia* of Rhamnaceae Hovenia dulcis Thunb, Hovenia acerba Lindl and Hovenia trichocarpa Chun et Tsiang. Furthermore, it is a common Medicine Food Homology (MFH) plant, which is abound in natural resources and widely distributed in China. More than 50% of *Semen Hoveniae* is crude fiber, and the flavonoids are the chief bioactive ingredients of *Semen Hoveniae*, with which 27.61% contents, including quercetin, myricetin, taxifolin and dihydromyricetin [1,2,3]. Extract of *Semen Hoveniae* had been proved to have multi-bioactivities, and the function of ameliorating alcohol-induced chronic liver damage was reported mostly at current research direction [4,5,6,7]. In fact, studies indicated that the hypoglycemic effects of the bioactive ingredients of MFH include saponins, flavonoids, terpenoids, alkaloids, and polysaccharides. Saponins and flavonoids are the main bioactive components present in the MFH species [8,9]. Quantities reports indicated that quercetin, myricetin, taxifolin and dihydromyricetin have significant activities against diabetes [10,11,12,13,14]. For this reason, extract of *Semen Hoveniae* is also a potential option for assistant hypoglycemic in MFH. Antioxidant therapeutic strategies are the modern methods of diabetes treatment. Research proved that the oxidative stress caused by the imbalance between free radicals and antioxidants in the body would destroy the function of pancreatic islets β cell, which is also a common pathogenic factor of diabetes [15,16,17]. Wu DT et al. [18] utilized seven extraction methods to extract polyphenolic-protein-polysaccharide complexes (PPPs) from *Hovenia dulcis*, and bioactivities of PPPs including inhibition on α-amylase and α-glucosidase and antioxidant activity were reported, interestingly, quercetin, myricetin, taxifolin and dihydromyricetin were determined in PPPs. Previous studies indicated that chemical drugs have some side effects including hypoglycemia, gain weight and chemical toxicity. Natural ingredients from Medicine Food Homology (MFH) as chemopreventive reagents against diabetes is one of the priorities [19]. Therefore, it makes sense to put focus on extraction optimization and antioxidant bioactivities of the four flavonoids from *Semen Hoveniae*. In addition, the total flavonoids in *Semen Hoveniae* were central research objects, which were not enough [20,21,22].

Much research on extraction methods of flavonoids has been reported, which focus on extraction solvent selection, extraction auxiliary method and so on. Each method has its strong and weak points. Wang L et al. [23] compared with soxhlet extraction and ultrasound-assisted extraction, and the extraction rates of total flavonoids were 0.47% and 0.65%, which showed that ultrasound-assisted extraction was superior to soxhlet extraction. Zhang HX et al. [24] compared with water and ethanol as solvent for total flavonoids extraction of *Semen Hoveniae*, the results have shown that extraction rates of 50% ethanol was 39.65% higher than of water. Wang YH et al. [14] summarized flavonoids extraction methods from *Semen Hoveniae,* including solvents extraction, heat reflux extraction, microwave-assisted extraction and ultrasound-assisted extraction, to find that the last two methods better than others. On the whole, ultrasound-assisted ethanol extraction maybe a preferred choice, with its efficient, environmental and recyclable features.

However, it is often a synergistic effect of various functional components on human health. The optimization of extraction with a variety of key flavonoids as a comprehensive indicator is significant enough to improve the efficacy value. Cai XJ et al. [25] optimized extraction process of saponin, total flavone and dry extract yield from *Semen Hoveniae* combining RSM with multi-index comprehensive evaluation method. Sun YF et al. [26] optimized extraction process of *Jinmai Gargle* based on analytic hierarchy process (AHP) and multi-index orthogonal design. The optimum extraction conditions were as follows: soaking for 30 min, adding 8 times of water, and decocting for 3 times with 30 min for each time, and the optimized extraction process is scientific, reasonable and stable. Han YF et al. [27] optimized the extraction process for *Yangyin Runmu* granules by Response Surface Methodology (RSM) based on entropy weight method-analytic hierarchy process method. The optimal extraction process condition was determined as adding 10 times of water, extracting 3 times, each time for 65 min. Lan JL et al. [28] optimized the extraction parameters of *Fengyin Decoction* with multi-index based on BAS-GA-BP neural network. The result was slightly better than that from the orthogonal test. The AHP compares the relative importance of the two factors to solve the multi-factor problem, and subdivides the complex problem into different levels, so as to build a scientific and accurate analytic hierarchy process index system and obtain satisfactory decision-making [29]. RSM is reliable for optimizing extraction process which widely used mathematical and statistical method to model and analyze process in which the response value is affected by variables [30,31].

As is known to all, the bioactive ingredients need to go through the gastrointestinal digestion process after entering the human body, and many effective ingredients are difficult to retain after digestion. It is absolutely necessary to study changes of bioactive constituent before and after digestion. In vitro simulated digestion is often lower cost, simpler and ethically uncontroversial, compared with in vivo digestion. In vitro simulated digestion includes static and dynamic digestion. Compared with static digestion, dynamic digestion is more comprehensively and truly, but more complex and costly. In vitro simulated digestion scheme should be niche-targeting, operable and effective considering the characteristics of test samples. Much research has shown that static simulated digestion model is still the most widely used. The final products of digestion are often affected by different parameter selection in the process of digestion, such as different sources of digestive enzymes, pH values, salt concentration, digestion time and so on, which the results are obvious different [32,33]. In this study, the composition of test object is relatively simple, and an international consensual standard model is adopted with comprehensive consideration, to make the results comparable and referential.

The evaluation of in vitro digestion of total flavonoids and total phenols from different food sources has been reported in a large number of literatures. The evaluation indicators usually include stability, antioxidant activity, anti-value-added activity, etc., but few studies have taken one or several single active substances as the research object, especially the study of comparing the monomer standard products. The current research shows that the content of flavonoids and phenols and the changes of antioxidant activity before and after simulated in vitro digestion are significantly different between natural foods and extracts from natural foods. Phenols and flavonoids are released from natural foods after in vitro digestion, so the content is significantly increased, and the antioxidant activity is significantly enhanced. The result of extract from natural food after digestion is opposite [34,35,36,37]. Wen X. et al. [38] compared the bioactivity of wild pink bayberry after in vitro digestion with that of bayberry extract, and found that the latter had higher antioxidant activity. In summary, food matrix can protect the structure of active substances such as flavonoids and polyphenols from being destroyed during gastrointestinal digestion, but the antioxidant activity of chyme after digestion is still significantly lower than that of natural ingredient extract, so it is necessary to develop efficient and green extraction methods and advanced technologies to improve the bioavailability of natural ingredient extract.

The present study had focused on four flavonoids including quercetin, myricetin, taxifolin and dihydromyricetin, and the process with ultrasound-assisted ethanol extraction of which was optimized based on AHP and multi-index comprehensive evaluation to improve the extraction rate. Next, the mimical in vitro gastrointestinal digestion test was performed to study the effect of digestion on four main flavonoids in the extracts of *Semen Hoveniae*. The study aims to provide reliable data to support optimal extraction process of *Semen Hoveniae* based on bioactivities, and further explore its biological utilization value, which can be applied to the functional food processing. The illustrative design of the present research is displayed as shown in Figure 1.

## 2. Materials and Methods

### 2.1. Materials

*Semen Hoveniae* used in the experiments were obtained in the city of Nanyang, Henan, China. Ethanol (purity ˃ 95%), aceti cacid (purity ˃ 98%), methanol (purity ˃ 98%), phenol (purity ˃ 98%) and acetonitrile (purity ˃ 98%) were provided by Sinopharm Chemical Reagent Co., Ltd. (Shanghai, China). Dihydromyricetin (purity ˃ 98%), taxifolin (purity ˃ 98%), myricetin (purity ˃ 98%), quercetin (purity ˃ 98%), were provided by Shanghai Yuanye Bio-Technology. Co., Ltd. (Shanghai, China). Dimethyl sulfoxide (purity ˃ 98%) was provided by Solarbio (Beijing, China). Pepsin, trypsin and bile salts were provided by Sigma (Saint Louis, MO, USA).

### 2.2. Extraction of Flavonoids Compounds

*Semen Hoveniae* were crushed with Chinese herbal medicine pulverizer and passing 40 mesh sieve plate. 1.0 g of the solid powder were extracted with different volume of ethanol in different concentrations, followed by ultrasound at different times (Table 1). Next, the mixture was centrifuged at 5 min. Take 1.0 mL of supernatant, filtered through a 0.22 μm (Nylon) syringe filter prior to analysis.

### 2.3. Experimental Design

This study used the method of Alberti et al. [39] with minor adjustments and the methods description partly reproduces their wording. Box and Behnken Design (BBD) was used to evaluate and optimize the extraction parameters. The effect of the independent variable extraction Solid-liquid ratio (*m*/*v*), A, the concentration of the solvent (*v*/*v*), B, and extraction time (min), C, were investigated in the extraction process at three variation levels (Table 1). In this case, 17 experiments were conducted to analyze the response pattern and to establish the predictable models for four flavonoids extraction.

The second-order polynomial model used in response surface analysis is shown in Equation (1), which is used to fit the experimental data of the research variables (1):(1)$$Y=\beta_{0}+\overset{3}{\underset{i=1}{\sum }}\beta_{i}X_{i}+\overset{3}{\underset{i=1}{\sum }}\beta_{\mathrm{ii}}X_{i}^{2}+\overset{2}{\underset{i=1}{\sum }}\overset{3}{\underset{j=i+1}{\sum }}\beta_{\mathrm{ij}}X_{i}X_{j}$$
where Y is the predicted response, β_0_, β_i_, β_ii_ and β_ij_ are the regression coefficients for intercept, linear, quadratic and interaction terms, respectively, X_i_ and X_j_ are the independent variables. The analysis of variance was used to check-out the statistical significance of the terms for each response. Non-significant terms were excluded from the initial model, and the significant parameters (*p* >= 0.05) rematched with the experimental data. The optimized parameters were further applied to validate the model with the same experimental procedure, and comparing theoretical predicted data with the experimental data to validate the prediction power of the models. Triplicate samples of the optimized proportion were prepared and analyzed.

### 2.4. Multi-Index Grading Method

Analytic Hierarchy Process (AHP) can be used to calculate the weight coefficient of each component in the multi index evaluation system to obtain the comprehensive evaluation value (Y) [40]. With reference to Jin Ying et al. [41], AHP multi index weighting method, the content of dihydromyricetin (A_1_), taxifolin (A_2_), myricetin (A_3_) and quercetin (A_4_) in *Semen Hoveniae* was allocated with reasonable weight coefficients. First, the AHP can construct a priority judgment matrix for comparison by comparing the relative importance of two factors. The evaluation scale is shown in Table 2.

Second, calculating the initial weight coefficient of four factors (A_1_, A_2_, A_3_, A_4_) according to the Equation (2):(2)$$W_{i}^{\prime }=\sqrt[{n}]{a_{i1}a_{i2}a_{i3}\ldots a_{\mathrm{in}}}$$
where W’ is the initial weight coefficient, a is the scale comparing two factors, i is the factor. Weight coefficient can be normalized in Equation (3):(3)$$W_{i}=\frac{W_{i}^{\prime }}{\sum_{i=1}^{n}W_{i}^{\prime }}\left(i=1,2,3\ldots n\right)$$
where $W_{i}$ is the eigenvector of factor, n is the number of factors.

Finally, calculate the consistency check results.
(4)$$\mathrm{CI}=\frac{\lambda_{\max}-m}{m-1}$$
(5)$$\mathrm{CR}=\frac{\mathrm{CI}}{\mathrm{RI}}$$
where $\lambda_{\max}$ is the matrix maximum eigenvalue; m is the number of indexes; CI is the consistency index; RI is the mean random consistency index; CR is the Random consistency ratio. When CR ˂ 0.1, the matrix meets the requirements, otherwise the model needs to be revised [29].

### 2.5. HPLC Analysis of Flavonoids Compounds in Optimum Conditions

The test used the method of Jong Suk Park et al. [42] with minor adjustment. The HPLC apparatus was Agilent 1260. Separation was performed on a Symmetry C18 (4.6 × 250 mm, 5 μm) column (Waters, Milford, MA, USA) at 25 °C.

The mobile phase was composed of solvent A (acetonitrile) and solvent B (0.1% acetic acid *v*/*v*). The following gradient was applied:12%:78% (A:B) at 0 to 5 min, 30%:70% (A:B) at 5 to 20 min, 35%:65% (A:B) at 20 to 24 min, 90%:10% (A:B) at 24 to 28 min, 12%:78% (A:B) at 28 to 32 min. The flow rate was 1.0 mL/min. The four flavonoids were identified by comparing retention time with standards. The runs were monitored at 365 nm.

The detection limit and quantitation limit of the target were determined by 3 times and 10 times of signal to noise ratio, respectively. Using the whole-process spiking test method, and selecting high, medium and low concentrations of dihydromyricetin, taxifolin, myricetin and quercetin for six parallel tests (*n* = 6), finally calculating the recovery and precision for the method of verification.

### 2.6. Analysis of Total Phenolic and Flavonoids Content

Refer to Folin–Ciocalteu method [43] for the determination of total phenolic. With gallic acid as the standard, the content of total phenols was determined by Thermo Scientific Microplate Reader. Determine the absorbance of the reaction solution at 756 nm.

Refer to AlCl_3_–NaNO color method [44] for the determination of total flavonoids. With rutin as the standard, the content of total flavonoids was determined by Thermo Scientific Microplate Reader. Determine the absorbance of the reaction solution at 510 nm.

### 2.7. In Vitro Simulated Digestion

The study referred to the method of Minekus et al. [32,45,46].with minor adjustment. Briefly, to simulate gastric digestion, 3 mL (5 mg/mL) of *Semen Hoveniae* extraction mixed with simulated gastric fluid (pepsin- added: 2000 U/mL) at 1:1 (*v*/*v*). Next, pH was adjusted at 2 by adding 1.0 mol/L HCL. The mixture was incubated for 0.0, 0.5, 1.0, 1.5, 2.0 h at 37 °C at 100 r/min. Next, the gastric digest and the intestinal fluid were mixed at 1:1 (*v*/*v*), and the mixture incubated with bile salts and trypsin (trypsin- added:100 U/mL). The mixture was incubated at pH 7 for 0.0, 1.0, 2.0, 3.0 h at 37 °C at 100 r/min. After digestion, the supernatant was collected and deposited in the refrigerator at subzero 80 °C until further use. A control group was carried out in the exact same conditions as the gastrointestinal digestion, but without the addition of pepsin and trypsin.

### 2.8. Antioxidant Capacity Assay

The study refers to the DPPH method of Ouerfelli M et al. [47]. This method determines the capacity of the free radical scavenging. The extract and DPPH solution are mixed evenly in 1:1. The absorbance was measured at 517 nm by spectrophotometer while the solution had been allowed to stand in the dark for 30 min until stabilization. The amount of sample required to reduce the DPPH concentration to 50% can represent the value of antioxidant capacity, expressed by EC_50_. The lower the EC_50_, the higher the antioxidant power.

The study referred to the ORAC method of Roy MK et al. [48] with minor adjustment. The antioxidants can block peroxyl radical-mediated fluorescein oxidation during 30 min by determining the ratio of Ex485 nm/Em518 nm. The area under each curve of Ex485 nm/Em518 nm with Trolox additions during 30 min were calculated as the AUC. The Trolox (2.5, 12.5, 25, 50 and 100 μmol/L) were used to plot standard curve between AUC and Trolox concentrations. The ORAC activity of *Semen Hoveniae* extraction was interpolated AUC and was expressed as Trolox equivalents.

### 2.9. Statistical Analysis

SPSS (24.0) statistical analysis software (IBM, Chicago, IL, USA) was used for one-way ANOVA and Duncan multiple comparison method for data statistics. Origin 2019 (Stat-Ease Inc., Minneapolis, MN, USA) were used for graphing. All measurements were repeated for 3 times, and all variables had their variance analyzed using the F test (two groups) or by Hartley’s test (*p* ≥ 0.05).

## 3. Results and Discussion

### 3.1. Multi-index Grading Method with AHP

Carry out hierarchical analysis of indicators to determine the judgment matrix after comparison of two indicators, as shown in Table 3. that the result shows that CR = 0.0072, which means that the evaluation system is correct and reliable, and can be used for the design and evaluation of the extraction parameter optimization.

The composite score is shown in Equation (6):Y = 0.4668×_1_ + 0.2776×A_2_ + 0.1603×A_3_ + 0.0953×A_4_(6)

### 3.2. Methodology Validation for HPLC Analysis

The method was determined by comparing the retention time of the chromatogram of the mixed standard under the same analysis parameters. The chromatographic profiles of standard mixture were shown in Figure 2. Quantification was performed with calibration curves of standards (Table 4). The limit of detection(LOD) and limit of quantitation (LOQ) of dihydromyricetin, taxifolin, myricetin and quercetin were in the range of 0.2~0.3 μg/mL and 0.5~0.8 μg/mL. The relative standard deviation (RSD) values were both below 4.67%, and the average recoveries were 83.4~110.6% for all analytes. These results conform to the technical standards and are sufficient to prove the reliability of the method referring to GB27404-2008 [49]. The results are summarized in Table 5, and Chromatographic profiles of analytes are shown in Figure 3.

### 3.3. Optimization of Semen Hoveniae Flavonoids Extraction by BBD

On the basis of preliminary experiments, the extract parameters of independent variables were fixed as follows: Solid-liquid ratio (A:1:20, 1:30, 1:40 *w*/*v*), solvent concentration (B:50, 70, 90 *v*/*v*), extraction time (C:30, 50, 70 min). Analyzing the interaction between variables and optimal parameters with statistical model. The design matrix of 17 runs were depicted by employing BBD and their respective responses are presented with composite score (Table 1).

### 3.4. Fitting the Model

Three independent variables including Solid-liquid ratio, Solvent concentration, and extraction time were investigated and optimized individually using BBD (Table 1). The composite score of *Semen Hoveniae* four flavonoids ranged statistically (*p* < 0.05) from 3.94 (assay number 7) to 7.06 (central point). The highest values for composite score were observed at the central point of the experimental design with solid-liquid ratio 1:30 *w/v* and 70% ethanol for 50 min (central point). The multiple regression analysis of flavonoids extracts showed that the model was significant (*p* < 0.05), did not present lack of fit (*p* = 0.0688), and it could explain 88.50% of variance in data (R^2^_adj_ = 0.74) (Table 6).

The relationship between variables and independent variables is described by the three-dimensional response surface graph generated by the model (Figure 4). The shape of the contour map (ellipse or circle) reflects the significance of the relationship between the corresponding variables. All charts indicated that solvent concentration and quadratic regression coefficient of solvent concentration have significant negative effect on flavonoids extraction, and solid-liquid ratio significantly increased the flavonoid extraction. The extraction time, quadratic regression coefficient of time, solid-liquid ratio and interactions of each other were not notably influence. The model for the predicted yield of four flavonoids could be expressed by the following quadratic polynomial equations (Equation (7)):Y = 0.6958A−0.9591B−0.1682C + 0.4419AB + 0.5100AC + 0.1784BC−0.4101A^2^−1.39B^2^−0.5614C^2^(7)

### 3.5. Verification of Predictive Models

The model predicted that the best extraction parameters were 1:36.75 *w/v* and 67.90% ethanol for 44.48 min for this combination of variables. In order to facilitate operability, the extracted parameters are unified as 1: 37 *w/v* and 68% ethanol for 45 min The observed and predicted values, along with the computed relative error (RE) for extraction were: dihydromyricetin (mg/g) (observed: 13.46 ± 0.38, predicted: 13.69, RE = 1.68%), taxifolin (mg/g) (observed: 1.76 ± 0.03, predicted: 1.51, RE = 16.56%), myricetin (mg/g) (observed: 0.36 ± 0.01, predicted: 0.37, RE = 2.70%), quercetin (mg/g) (observed: 0.25 ± 0.006, predicted: 0.23, RE = 8.69%), and composite score (observed: 7.15 ± 0.19, predicted: 7.12, RE = 0.42%). Due to the low relative error values obtained by the comparison between observed and predicted values, the proposed model could be used to predict the response value. This result was almost consistent with H Zhang et al. [30].

### 3.6. Effect of Digestion on the Flavonoids and Total Phenolic Contents

The total phenolic was determined before and during the simulated digestion. The result was shown in Figure 5. As can be seen, during gastric digestion, the total phenol content decreased slightly (with pepsin: decreased to 84.6% of the initial value, without pepsin: decreased to 89.78% of the initial value). There was no significant difference between the pepsin-added group and the control group. During intestinal digestion, there was a very significant difference between the trypsin-added group and the control group. The total phenol content of the trypsin-added group increased slightly compared with the initial value. On the contrary, the total phenol content of the control group decreased significantly after digestion in 2 h.

We guess that phenols are relatively stable in acidic environment, which is consistent with most previous reports [50,51]. The intestinal tract is in a weak alkaline environment, which is not conducive to the stability of phenolic compounds. In the extraction solution, phenols, proteins and polysaccharides often form relatively stable complexes with hydrogen bonds and hydrophobic bonds, which are released with the action of acids, bases and enzymes. At the same time, the synergism of trypsin and bile salt increased the stability of phenols in digestive fluid, and new phenols were detected by UV [52]. This is consistent with the research results of Cong Y.L. et al. [53], which found that the release of total phenols in the intestinal digestion stage of citrus increased during in vitro simulated gastrointestinal digestion. The phenols in the reaction system non- enzyme-added were significantly decreased by pH.

The total flavonoids were determined before and during the simulated digestion. The result was shown in Figure 6. As can be seen, the content tendency of total flavonoids is similar to that of total phenols in gastrointestinal digestion. During gastric digestion, the total flavonoids content decreased slightly (with pepsin: decreased to 79.96% of the initial value, without pepsin: decreased to 70.34% of the initial value). During intestinal digestion, the total flavonoids content decreased slightly to 78.43% of the initial value with trypsin group, while decreased to 54.33% of the initial value without trypsin group. In addition, the theory should be similar to that of total phenol.

Dihydromyricetin standard was determined before and during the simulated digestion. The result was shown in Figure 7. As can be seen, digestion had an influence on dihydromyricetin content, mainly for intestinal phases, where significantly reduced the dihydromyricetin content compared to the crude (undigested) extract (below the detection limits). Interestingly, there was no significant difference between the dihydromyricetin content of the extract digested in the gastric phase and the crude extract. That is to say, dihydromyricetin was not significantly affected by pepsin or trypsin, and pH was possibly an essential interfering factor.

Dihydromyricetin, taxifolin, myricetin and quercetin were investigated in *Semen Hoveniae* extracts before and during the simulated digestion, and the taxifolin were the class found in higher concentrations (Table 7). Before and during simulated digestion, the flavonoids compounds: dihydromyricetin, myricetin and quercetin were below the detection limits, and taxifolin content was declined to 34.87% (experimental group) and 44.67% (control group).

Following the gastric digestion, in comparison to undigested extracts, the content of two flavonoids compounds (dihydromyricetin, taxifolin) had no significant changes (*p* > 0.05), and myricetin and quercetin were significantly decreased in extracts (*p* < 0.05). In order of stability during digestion, the four flavonoids compounds were dihydromyricetin > taxifolin > myricetin > quercetin.

After the intestinal phase, compared to the gastric digestion, a significant (*p* < 0.05) decline was observed for the four compounds in *Semen Hoveniae* extracts. Dihydromyricetin, myricetin and quercetin were below detection limits. Taxifolin was retained about 34.87~44.67%.

In summary, when the extract is digested through gastrointestinal tract, the phenolic and flavonoid are greatly affected by pH and digestive enzymes. It is relatively stable under acidic conditions, and the chemical structure of a single substance is extremely easy to be destroyed in an alkaline environment, and then forms a new structure of phenols or flavonoids or other substances. The content of total phenols and total flavonoids may basically maintain a dynamic balance, but it is difficult to retain a single substance.

### 3.7. Effect of Digestion on the Antioxidant Capacity

The research shows that different chemical methods of antioxidant activity have poor correlation. So far, there has been no standard method for evaluating the antioxidant activity of natural products accurately and comprehensively. Therefore, at least two methods need to be used for comprehensive evaluation at the same time, since each method has respective reaction mechanism, which reflects different antioxidant capacity. [54]. According to the reaction mechanism, chemical assays can be divided into hydrogen atom transfer (HAT) and single electron transfer (SET). The Oxygen Radical Absorption Capacity (ORAC) mechanism is that antioxidants provide hydrogen atoms to neutralize oxygen free radicals and protect fluorescein from oxygen free radicals. Furthermore, the antioxidants are stronger than that of protected molecules in hydrogen supply ability. Therefore, the stronger the fluorescence intensity, the stronger the antioxidant activity of the natural product in ORAC assay. It belongs to the HAT mechanism typically. The DPPH (2,2-di(4-tert-octylphenyl)-1-picrylhydrazyl) assay is that antioxidant provide electron to neutralize DPPH radical. It is to evaluate the free radical scavenging ability [55].

The antioxidant capacity assessed by DPPH and ORAC assays of *Semen Hoveniae* extracts before and during the simulated digestion is shown in Figure 8. After gastric digestion, the DPPH values of all samples did not significantly (*p* > 0.05) change, compared with the initial DPPH values (pepsin-added: from 263.72 μmol/L to 256.30 μmol/L, control group: from 248.76 μmol/L to 256.30 μmol/L). A significant (*p* < 0.05) decrease in antioxidant capacity for all samples after the intestinal digestion was observed in comparison with the gastric DPPH values, and the initial DPPH values (TEAC of two group was zero after one hour). These results are in agreement with Cedola A et al. [56] who also observed a significant reduction in antioxidant capacity by the DPPH method for oleuropein of olive leaf extracts subjected to in vitro intestinal digestion. Some correlations between flavonoids and antioxidant activity were found after comparing changes before and during the simulated digestion. These results are in agreement with Ge H et al. [57] who also observed correlations between flavonoids and antioxidant activity for Rhizoma Chuanxiong extracts subjected to in vitro intestinal digestion.

For the ORAC assay, the antioxidant capacity behaved statistically different (*p* < 0.05). Following the gastric phase, a variable behavior in the ORAC values was observed. In comparison to the initial ORAC values, a significant (*p* < 0.05) increase in ORAC values was verified for the extraction (125.52%,116.73% of initial values). After intestinal digestion, without trypsin group showed a significant (*p* < 0.05) decrease in the ORAC values, in comparison to the gastric and initial ORAC values (60.02% of initial values), while trypsin-added group showed a significant (*p* < 0.05) increase in the ORAC values (from 8007.34 μmol/L to 9824.52 μmol/L). These results proved a positive effect of trypsin on antioxidant capacity.

In conclusion, the DPPH assay showed no antioxidant activity after gastrointestinal digestion, but the ORAC assay displayed a opposite result and even antioxidant activity increased in trypsin-added group. It is speculated that flavonoids are hydrolyzed into small molecular substances by trypsin, and the covered hydroxyl groups are exposed, or compounds containing more hydroxyl groups are generated, which significantly enhance the ability of hydrogen donor atoms, while the ability of electron transfer is weakened, which can also explain why the antioxidation activity tested by DPPH method is significantly reduced. Another conjecture is that Pepsin is hydrolyzed by trypsin and the hydrolysate has certain ORAC antioxidant activity [58].

## 4. Conclusions

The extraction parameters of the four typical flavonoids from *Semen Hoveniae* were optimized by multi-index comprehensive evaluation based on AHP. The results revealed that three influence factors acted significantly and the order was: ethanol concentration > solid-liquid ratio > ultrasound time. The optimized extraction parameters were as follows: 1:37 *w*/*v* of solid-liquid ratio, 68% of ethanol concentration and 45 min for ultrasonic time. The composite score was 7.13 and the best predicted score was 7.15, while the model was stable and reliable. Four flavonoids in extracts were mostly degraded into other bioactive substances after in vitro gastrointestinal digestion, while the total phenolic and flavonoids had a good retention, which may have stronger antioxidant activity by ORAC but not by DPPH assay.

On the one hand, the results suggest that a single effective ingredient is difficult to be retained after digestion by the human body, and often forms other substances, with poor biological acceptability. On the other hand, it reveals the interaction between the mixed extracts, which is sometimes conducive to the conversion of the effective ingredients into other effective ingredients, and then play its role. This implies that when extracting effective ingredients from natural products, it may not be that the higher the purity of the extracted single ingredient, the better the bioavailability.

At present, we are not clear about the specific mechanism of high ORAC antioxidant value after gastrointestinal digestion. Is it the role of new compounds formed by the interaction of the mixture or the role of some special compounds degraded into compounds with a similar structure? These contents need to be further explored. Based on the underlying principle of exerting antioxidant activity, compounds with similar activity or structure are gathered and then extracted by comprehensive scoring method, which may greatly help to improve bioavailability.

This study initially combines the extraction optimization of comprehensive score of multi-effective substances with the evaluation of in vitro activity, to provide a new idea for further exploring the methods to improve the bioavailability of effective ingredients, and throw away a brick in order to acquire a gem for the study of the mechanism and principle of the in vivo effective ingredients.

## Figures and Tables

**Figure 1 foods-12-00773-f001:**
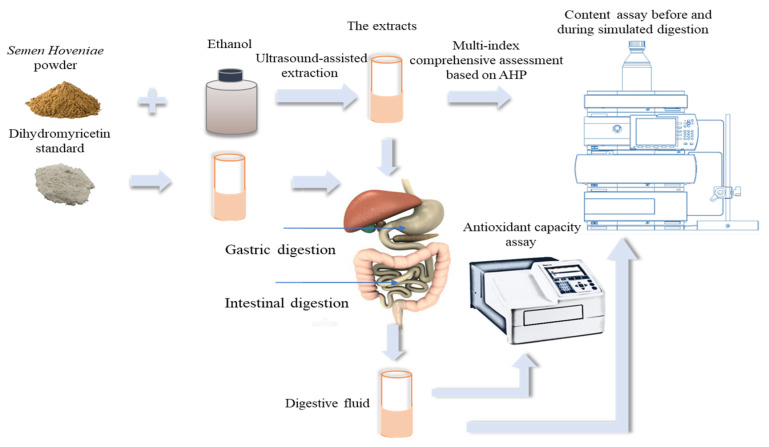
The illustrative design of the present research. Including three sections: optimized extraction process, simulated gastrointestinal digestion and evaluated of extracts, simulated gastrointestinal digestion and evaluated of the dihydromyricetin standard.

**Figure 2 foods-12-00773-f002:**
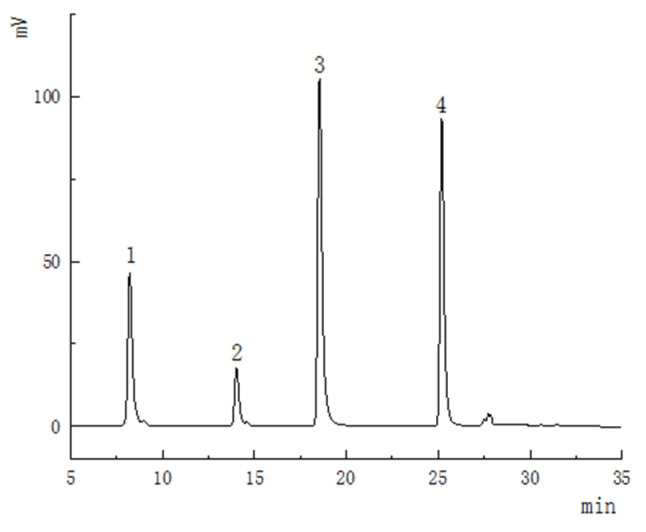
Chromatographic profiles of standard mixture. 1: Dihydromyricetin; 2: Taxifolin; 3: Myricetin; 4: Quercetin.

**Figure 3 foods-12-00773-f003:**
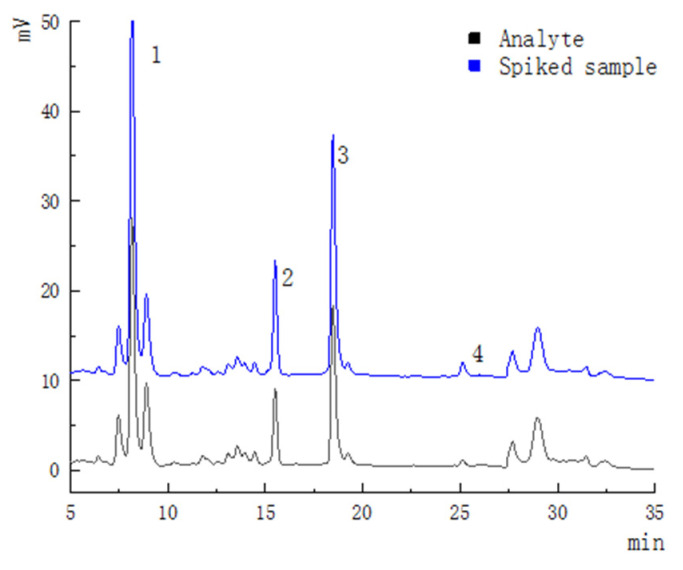
Chromatographic profiles of analytes and spiked sample. 1: Dihydromyricetin; 2: Taxi folin; 3: Myricetin; 4: Quercetin.

**Figure 4 foods-12-00773-f004:**
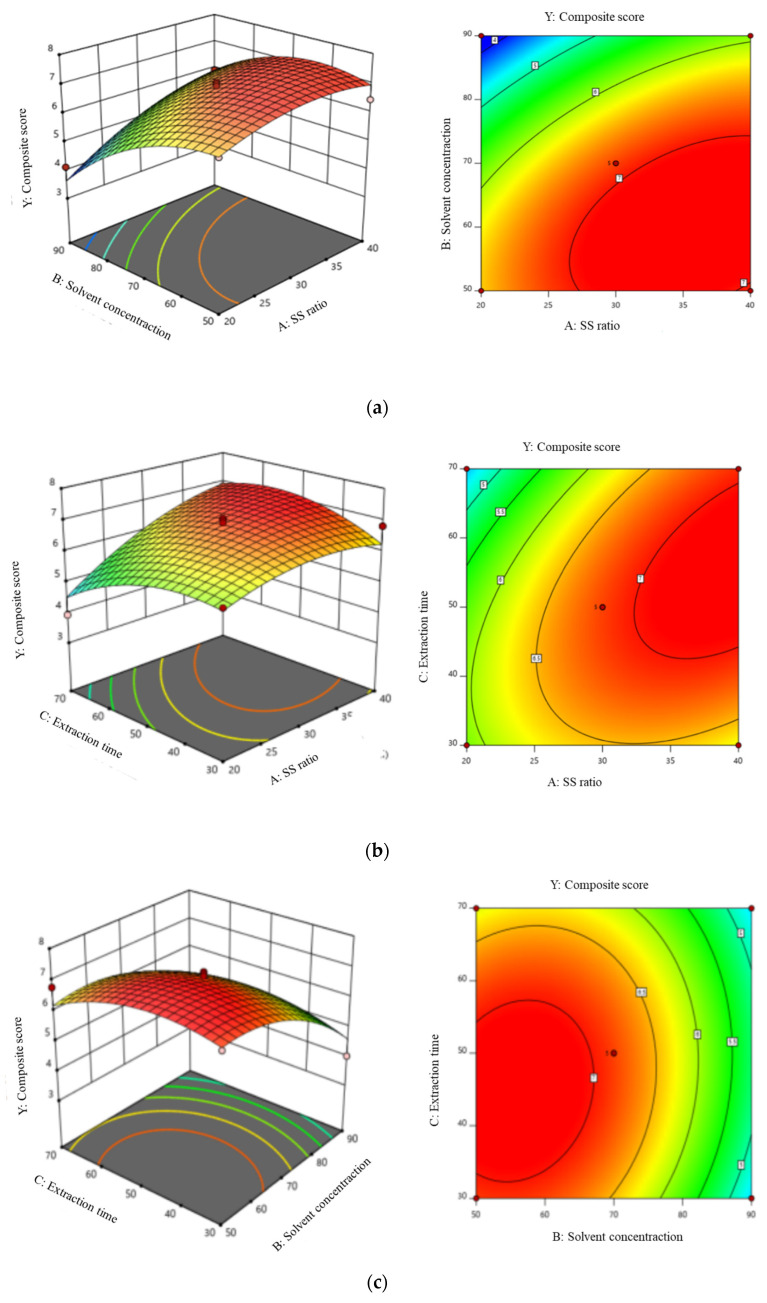
Response surface and counter plots showing the interaction effect on the extraction yield of flavonoids. (**a**): response surface graph of solvent concentration(B) and SS ratio(A); (**b**): response surface graph of extraction time(C) and SS ratio(A); (**c**): response surface graph of solvent concentration(B) and extraction time(C).

**Figure 5 foods-12-00773-f005:**
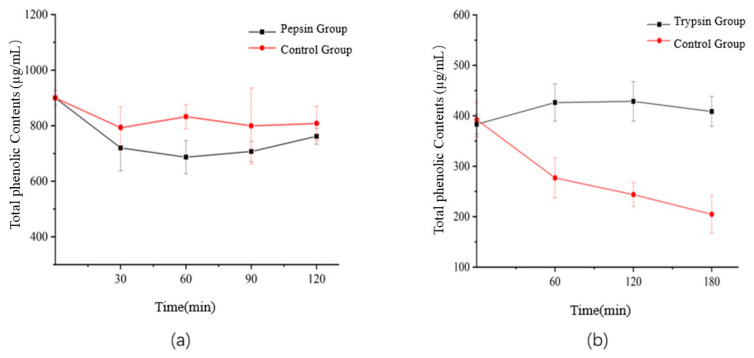
Changes of total phenols in simulated gastric (**a**) and intestinal (**b**) digestion of *Semen Hoveniae* in vitro.

**Figure 6 foods-12-00773-f006:**
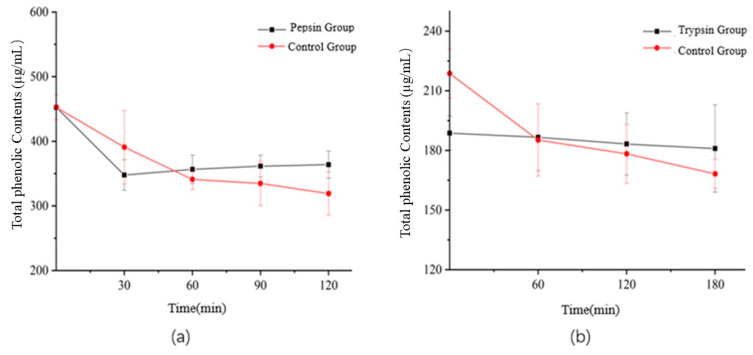
Changes of total flavonoids in simulated gastric (**a**) and intestinal (**b**) digestion of *Semen Hoveniae* in vitro.

**Figure 7 foods-12-00773-f007:**
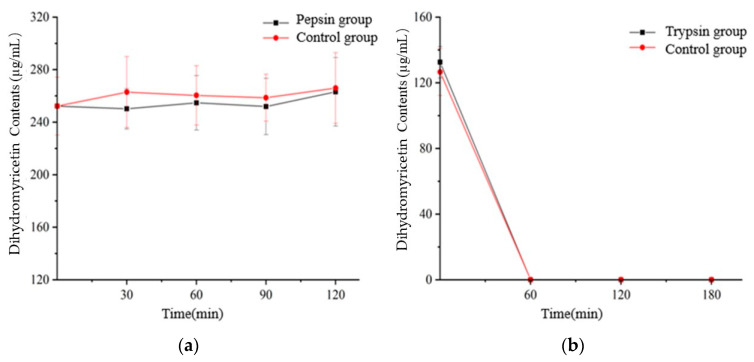
Changes of dihydromyricetin standard content of the extract digested before and during gastric phase (**a**) and intestinal phases (**b**).

**Figure 8 foods-12-00773-f008:**
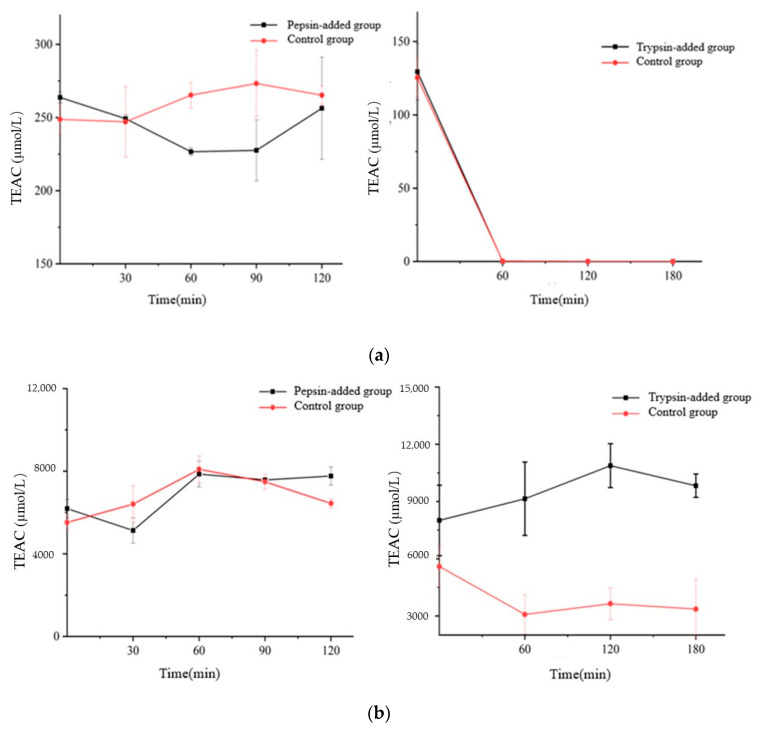
Antioxidant capacity from *Semen Hoveniae* extracts before and after simulated digestion in vitro. (**a**): DPPH assay. (**b**): ORAC assay.

**Table 1 foods-12-00773-t001:** BBD applied for flavonoids extracts from *Semen Hoveniae*.

Run	Factors			
	A	B	C	Composite Score
1	−1	−1	0	6.42
2	+1	−1	0	6.44
3	−1	+1	0	4.13
4	+1	+1	0	5.92
5	−1	0	−1	5.96
6	+1	0	−1	6.82
7	−1	0	+1	3.94
8	+1	0	+1	6.83
9	0	−1	−1	6.80
10	0	+1	−1	4.02
11	0	−1	+1	6.78
12	0	+1	+1	4.71
13	0	0	0	7.05
14	0	0	0	6.26
15	0	0	0	7.06
16	0	0	0	7.00
17	0	0	0	6.93
True values				
−1	1:20	50	30	
0	1:30	70	50	
+1	1:40	90	70	

“−1” is the low level, “0” is the mean level, “+1” is the high level.

**Table 2 foods-12-00773-t002:** Interpretation of 1–9 scale method.

Judgment Scale	Definition	Reciprocal
1	The factor i is equally important than the factor j.	If the ratio of importance of factor i to factor j is dij, then the ratio of importance of factor j to factor i is dij = 1/dij
3	The factor i is more important than the factor j	
5	The factor i is obviously more important than the factor j	
7	The factor i is significantly more important than the factor j	
9	The factor i is extremely important than the factor j	
2,4,6,8	In the middle of the above two adjacent judgment scales	

**Table 3 foods-12-00773-t003:** Judgment matrix for the weights of indicators.

Project	A_1_	A_2_	A_3_	A_4_	W’	W
A_1_	1	2	3	4	2.2134	0.4668
A_2_	1/2	1	2	3	1.3161	0.2776
A_3_	1/3	1/2	1	2	0.7598	0.1603
A_4_	1/4	1/3	1/2	1	0.4518	0.0953
Consistency Check		CR = 0.0072 < 0.1			Pass	

**Table 4 foods-12-00773-t004:** Standard curve equation detection.

Four Flavonoids	Regression Equation	Linear Range (μg/mL)	R^2^	LOD (μg/mL)	LOQ (μg/mL)
Dihydromyricetin	Y = 0.5650X − 25.0060	10~2000	0.9919	0.3	0.8
Taxifolin	Y = 0.82380X + 4.2838	10~1000	0.9992	0.3	0.6
Myricetin	Y = 31.1250X − 233.3900	10~1000	0.9989	0.2	0.5
Quercetin	Y = 33.6970X − 271.1300	1~250	0.9903	0.2	0.5

**Table 5 foods-12-00773-t005:** Data for recovery and RSD studies of dihydromyricetin, taxifolin, myricetin and quercetin.

Analyte	Nominal Concentration (μg/mL)	Observed Concentration (μg/mL, *n* = 6)	Mean Recovery (%)	RSD (%)
Dihydromyricetin	50	47.87 ± 0.89	95.74	3.55
	200	184.33 ± 0.47	92.16	1.39
	800	756.9 ± 0.25	94.61	3.91
Taxifolin	10	11.56 ± 0.15	110.6	4.67
	40	36.46 ± 0.09	91.15	1.04
	160	140.1 ± 0.17	87.5	2.13
Myricetin	10	10.36 ± 0.13	103.6	0.97
	40	42.59 ± 0.15	106.5	2.53
	160	134.57 ± 0.23	84.11	1.55
Quercetin	5	4.17 ± 0.06	83.4	1.22
	20	19.03 ± 0.12	95.15	3.31
	80	77.87 ± 0.14	97.34	0.57

**Table 6 foods-12-00773-t006:** Analysis of variance for the fitted quadratic polynomial model for recovery of composite score.

Source	SS	df	MS	F-Value	*p*-Value
Model	18.08	9	2.01	5.99	0.0139 *
A-Solid-liquid ratio	3.87	1	3.87	11.54	0.0115 *
B-Solvent concentration	7.36	1	7.36	21.93	0.0023 **
C-extration time	0.23	1	0.23	0.67	0.4387
AB	0.78	1	0.78	2.33	0.1709
AC	1.04	1	1.04	3.1	0.1217
BC	0.13	1	0.13	0.38	0.5575
A^2^	0.71	1	0.78	2.11	0.1896
B^2^	2.18	1	2.18	6.49	0.0382 *
C^2^	1.33	1	1.33	3.95	0.0871
Residual	2.35	7	0.34		
Lack of Fit	1.88	3	0.63	5.38	0.0688
Pure Error	0.47	4	0.12		
Cor Total	20.43	16			

* Significant at *p*-value < 0.05. ** Significant at *p*-value < 0.01.

**Table 7 foods-12-00773-t007:** Content of individual flavonoids compounds in *Semen Hoveniae* extracts before and during the simulated digestion.

Flavonoids Compounds	Group	Undigested (%)	Gastric Phase (%)	Intestinal Phase (%)
Dihydromyricetin	Enzyme-added	100	94.39 ^a^	0 ^b^
	Eontrol	100	101.45 ^a^	0 ^c^
Taxifolin	Enzyme-added	100	84.18 ^a^	34.87 ^b^
	Control	100	77.57 ^b^	44.67 ^c^
Myricetin	Enzyme-added	100	75.84 ^b^	0 ^b^
	Control	100	77.99 ^b^	0 ^c^
Quercetin	Enzyme-added	100	67.75 ^b^	0 ^c^
	Control	100	72.94 ^b^	0 ^c^

a,b,c different letters indicate significant difference (*p* < 0.05).

## Data Availability

Data is contained within the article.

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
