# Peer review of "Multi-Index Comprehensive Assessment Optimized Critical Flavonoids Extraction from Semen Hoveniae and Their In Vitro Digestive Behavior Evaluation"

_foods, 2023, doi:10.3390/foods12040773_

Round 1

Reviewer 1 Report

·         What s the brand of Semen Hoveniae material?

·         The words in figure 1 can t be read, so they should be rearranged.

·         At table 3, the first letters of myricetin and quercetin should be Myricetin and Quercetin.

·         At table 5, Residua should be written as Residual

·         DPPH and ORAC results should be supported with literature.

Reviewer 2 Report

I reviewed the manuscript entitled, multi-index comprehensive assessment based on AHP theory optimized critical flavonoids extraction from Semen Hoveniae and their in vitro digestive behavior evaluation. The title must be revised for easy understanding. There is no novelty in performing work. Authors optimized the extraction of flavonoids. This study has many experimental errors and many methods are old. Study approach is not a novel. Based on all these observations, I must recommend rejection.

What is the meaning of critical flavonoids extraction?

Authors should identify the total flavonoid content

There are many studies available in literature on extraction and optimization studies.

If the authors aimed at flavonoids, extracts must be purified. Performing in vitro digestion without purification may yield less bioavailability

There are no studies related to bioavailability and bioaccessibility

Evaluation can be removed from the title

Line 6: remove affiliation 1

Line 9: remove they are co-first authors

What is the meaning of critical flavonoids?

Figure 1 quality I extremely poor

There is no ref for section 2.5

Results and discussion are poorly written and failed to exclusively compare with available literature

Conclusions must be revised to reflect the content described

References are not according to the journal format 

Reviewer 3 Report

Multi-index comprehensive assessment based on AHP theory optimized critical flavonoids extraction from Semen Hoveniae and their in vitro digestive behavior evaluation

·       Please specify the correct plant name “Hoveniae Semen / Semen Hoveniae” and the name should be italics or not?

·       What is the part of used? Seed / Radix ?

·       What is the species of Hovenia? Hovenia dulcis Thunb., Hovenia acerba Lindl. and Hovenia trichocarpa Chun et Tsianga.

·       State the reason for selecting the quantitative analysis of dihydromyricetin, taxifolin, myricetin and quercetin. Why is it not analyzing the dihydroquercetin? The four main bioactive flavonoids are dihydromyricetin, dihydroquercetin, myricetin, quercetin. This plant has a wide range of biological activity and has many active substances.

·       Add references for performing “In Vitro Simulated Digestion”.

·       What is the relationship between the hypoglycemic effect of researchers’ interest that indicated in the Introduction and the antioxidant activities done in the experiment?

·       Describe the different mechanisms between ORC and DPPH used in this study.

·       The hypothesis of the mechanism by which enzymes (pepsin and trypsin) affect the structure of flavonoids and metabolites with their antioxidant activities (ORAC) should be presented.

·       There are various methods for in vitro digestion protocols. Why did the researcher choose to do only certain topics?

·       Provide reference documentation for HPLC quantification method. Show the chromatograms (extracts and solution obtained from digestion), suitability result (%RSD, resolution, theoretical plates, symmetry and purity of peaks), LOD, LOQ of the method in supplementary materials.

·       How is “Multi-index comprehensive assessment based on Analytic Hierarchy Process (AHP) method” better than other classical methods for this study?

Reviewer 4 Report

The manuscript entitled "Multi-index comprehensive assessment based on AHP theory optimized critical flavonoids extraction from Semen Hoveniae and their in vitro digestive behavior evaluation" has been reviewed and the following comments should be considered by author for improvement the manuscript :

-The title is quite long and must not contain abbreviations " the title should be concise and informative".

-The scientific name of the plant must be presented in the introduction "which one of these species? Hovenia acerba (syn Hovenia kiukiangensis), Hovenia dulcis Thunb. syn Hovenia inaequalis DC, Hovenia parviflora, Hovenia pubescens, Hovenia robusta, Hovenia tomentella, Hovenia trichocarpa.

-Extensive editing of English language and style is required throughout the manuscript 

-Line 218 "Solid-liquid ratio (A:1:20, 1:30, 1:40 g/mL)", it is better to expressed as "(w/v)"

-In the results and discussion section the extraction optimization conditions are predicted by RSM but there is no results for real experiments based on the RSM suggested conditions.

-The results of total phenolic and flavonoids content for every suggested variant, must be added and interpreted for every suggested conditions

-The HPLC flavonoids profile of the extract must be added and interpreted or every suggested conditions

-The antioxidant capacity by DPPH and ORAC  results must be added and interpreted for every suggested conditions

and finally the experimental results will compare with the suggested conditions to confirm the suggested optimal extraction conditions

-The conclusion should be improved to focus on the novelty which must be supported with the results, the potential application and limitation must be added as well.

Round 2

Reviewer 2 Report

Authors reply to my comments are convincing. In my opinion, this version can be accepted for publication.

Author Response

Thank you very much for your recognition

Reviewer 3 Report

Multi-index comprehensive assessment optimized critical flavonoids extraction from Semen Hoveniae and their in vitro digestive behavior evaluation

English language used to edit the manuscript should be extensive revised.

The introduction did not provide sufficient background and include all relevant references. It should be mentioned the previous studies about the gastrointestinal digestion of flavonoids and phenols.

No data of HPLC system suitability.

Cited the reference of criteria used to determine the validation method. The criteria should be showed in the manuscript.

The retention time of taxifolin was not consistent with that of standard taxifolin, according to chromatographic profiles of analytes shown in Figure S3. The specificity data should be showed.

All purity values of all compounds should be presented.

In Table S4, the lowest concentration of dihydromyricetin in linearity range was lower than LOQ. It’s not correct.

In Topic 3.6, the total phenolic was determined before and during the simulated digestion. How to verify that other substances in the medium do not interfere with the analysis results?

Shows a diagram of the chemical structure changes when the compounds are digested.

Reviewer 4 Report

The revised version has been reviewed and the found that the authors well considered all the reviewer comments which is improved the manuscript and made it suitable for publication in Foods.

Author Response

(The authors gave the same response as above.)
